# Inquiry of the Metabolic Traits in Relationship with Daily Magnesium Intake: Focus on Type 2 Diabetic Population

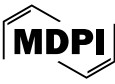

Ana-Maria Gheorghe [1,2], Mihai-Lucian Ciobica [3,4,*], Claudiu Nistor [5,6,*], Maria-Magdalena Gurzun [7,8], Bianca-Andreea Sandulescu [1,3,4], Mihaela Stanciu [9], Florina Ligia Popa [10] and Mara Carsote [2,11]

1   PhD Doctoral School, "Carol Davila" University of Medicine and Pharmacy, 020021 Bucharest, Romania; ana-maria.gheorghe@drd.umfcd.ro (A.-M.G.); bianca-andreea.sandulescu@drd.umfcd.ro (B.-A.S.)
2   Department of Clinical Endocrinology V, "C.I. Parhon" National Institute of Endocrinology, 011863 Bucharest, Romania; carsote_m@hotmail.com
3   Department of Internal Medicine and Gastroenterology, "Carol Davila" University of Medicine and Pharmacy, 020021 Bucharest, Romania
4   Department of Internal Medicine I and Rheumatology, "Dr. Carol Davila" Central Military University Emergency Hospital, 010825 Bucharest, Romania
5   Department 4-Cardio-Thoracic Pathology, Thoracic Surgery II Discipline, "Carol Davila" University of Medicine and Pharmacy, 050474 Bucharest, Romania
6   Thoracic Surgery Department, "Dr. Carol Davila" Central Military University Emergency Hospital, 010242 Bucharest, Romania
7   Cardiology Discipline, "Carol Davila" University of Medicine and Pharmacy, 020021 Bucharest, Romania; magdalena.gurzun@umfcd.ro
8   Laboratory of Non-Invasive Cardiovascular Exploration, "Dr. Carol Davila" Central Military University Emergency Hospital, 010242 Bucharest, Romania
9   Department of Endocrinology, Faculty of Medicine, Lucian Blaga University of Sibiu, 550024 Sibiu, Romania; mihaela.stanciu@ulbsibiu.ro
10  Department of Physical Medicine and Rehabilitation, Faculty of Medicine, "Lucian Blaga" University of Sibiu, 550024 Sibiu, Romania; florina-ligia.popa@ulbsibiu.ro
11  Department of Endocrinology, "Carol Davila" University of Medicine and Pharmacy, 020021 Bucharest, Romania
*   Correspondence: lucian.ciobica@umfcd.ro (M.-L.C.); claudiu.nistor@umfcd.ro (C.N.)

**Abstract:** Magnesium (Mg), an essential nutrient with a wide area of physiological roles, stands as a cofactor in over 600 enzymatic reactions involved in the synthesis of proteins and nucleic acids, DNA repair, neuromuscular functions, neuronal transmission, cardiac rhythm regulation, and the modulation of metabolic pathways, as well as acting as a natural blocker for the calcium channels. Our objective was to highlight the most recent clinical data with respect to daily Mg intake (DMI) and metabolic traits, particularly type 2 diabetes mellitus (DM). This was a PubMed-based review of the English-language medical papers across different key terms of search; the time frame was from January 2019 until April 2024. We included (clinically relevant) original studies and excluded cases reports, series, reviews, editorials, opinion, experimental studies, and non-human data as well as studies that did not specifically assessed DMI and only provided assays of serum Mg, studies on patients diagnosed with type 1 or secondary DM. A total of 30 studies were included and we organized the key findings into several sections as follows. Studies investigating DMI in relationship with the adherence to local recommendations in diabetic subjects (n = 2, one transversal and another retrospective cohort; N = 2823) found that most of them had lower DMI. Deficient DMI was correlated with the risk of developing/having DM across five studies (n = 5, one prospective and four of cross-sectional design; N = 47,166). An inverse correlation between DMI and DM prevalence was identified, but these data are presented amid a rather heterogeneous spectrum. Four novel studies (N = 7279) analysed the relationship between DMI and DM control according to various methods (HbA1c, fasting and postprandial glycaemia, and insulin); the association may be linear in diabetic subjects only at certain levels of DMI; additionally, the multifactorial influence on HBA1c should take into consideration this dietary determinant, as well, but there are no homogenous results. Three studies concerning DMI and diabetic complications (one cross-sectional, one prospective, and another case–control study) in terms of retinopathy (n = 1, N = 3794) and nephropathy (n = 2, N = 4805) suggested

a lower DMI was associated with a higher risk of such complications. Additionally, two other studies (one prospective and one retrospective cohort) focused on mortality (N = 6744), which, taking only certain mortality indicators into consideration, might be decreased in the subgroups with a higher DMI. Seven studies (N = 30,610) analysed the perspective of DMI in the general population with the endpoint of different features amid glucose profile, particularly, insulin resistance. Concerning HOMA-IR, there were three confirmatory studies and one non-confirmatory, while fasting plasma glucose was highlighted as inversely correlated with a DMI (n = 1). The highest level of evidence regarding Mg supplementation effects on glucose metabolism stands on seven randomised controlled trials (N = 350). However, the sample size was reduced (from 14 to 86 individuals per study, either diabetic or pre-diabetic) and outcomes were rather discordant. These clinical aspects are essential from a multidisciplinary perspective and further trials are mandatory to address the current areas of discordant results.

**Keywords:** diabetes mellitus; magnesium; metabolic syndrome; high blood pressure; daily intake; nutrient; glucose; glycated haemoglobin A1c

## 1. Introduction

Magnesium represents an essential nutrient with a wide area of physiological roles, including being a cofactor in over 600 enzymatic reactions involved in the synthesis of proteins and nucleic acids, DNA repair, neuromuscular functions, neuronal transmission, cardiac rhythm regulation, the modulation of metabolic pathways as well as acting as natural blocker for the calcium channels [1–5]. The European Food Safety Association (EFSA) recommends a daily magnesium intake of 350 mg for men, and 300 mg for women, while for US the recommendation is slightly higher (420 mg/day for males and 320 mg/day for females) [6,7]. In humans, the most important dietary sources include plant-based foods rich in magnesium (such as whole grains, green vegetables, nuts, seeds) in addition to sources with a lower magnesium content (for example, legumes, fruit, meat, fish, and dairy) [8]. When it comes to (oral) magnesium supplementation, it should be noted that the bioavailability of different supplements largely varies, with organic formulations having a higher bioavailability versus inorganic products [9]. In spite of a wide variety of dietary sources, magnesium deficiency is quite common in recent years, and it is linked to numerous conditions (with pathogenic loops being more or less understood so far) including cardiovascular disease, type 2 diabetes mellitus, gastrointestinal conditions, chronic kidney disease, and osteoarticular disease, including low bone mineral density and even some parathyroid conditions [10–18].

Serum magnesium is unreliable when it comes to estimating total body magnesium (representing less than 1%) as this nutrient is mainly stored intracellularly, in bones, muscles and viscera [19–21]. The relationship between the daily magnesium intake and its serum levels is regulated by complex mechanisms also including the balance between nutrient absorption and excretion, the adjustment of its bone storage, the interaction with parathyroid hormone (PTH) and calcium (as essential contributors of the mineral metabolism), and the equilibrium between intracellular and extracellular magnesium content. Magnesium-related physiological profile include a balance between the dietary intake, nutrient absorption at distal small intestinal level, respectively, its renal excretion and reabsorption versus micronutrient storage into the bones (representing 50–60% of the total body content) and soft tissues such as the muscle, etc. Additional regulation is provided by passive para-cellular and trans-cellular transport at cecal and colonic level. The renal excretion is regulated by the serum levels despite the fact than less than 1% is found her [22–25]. (Figure 1)

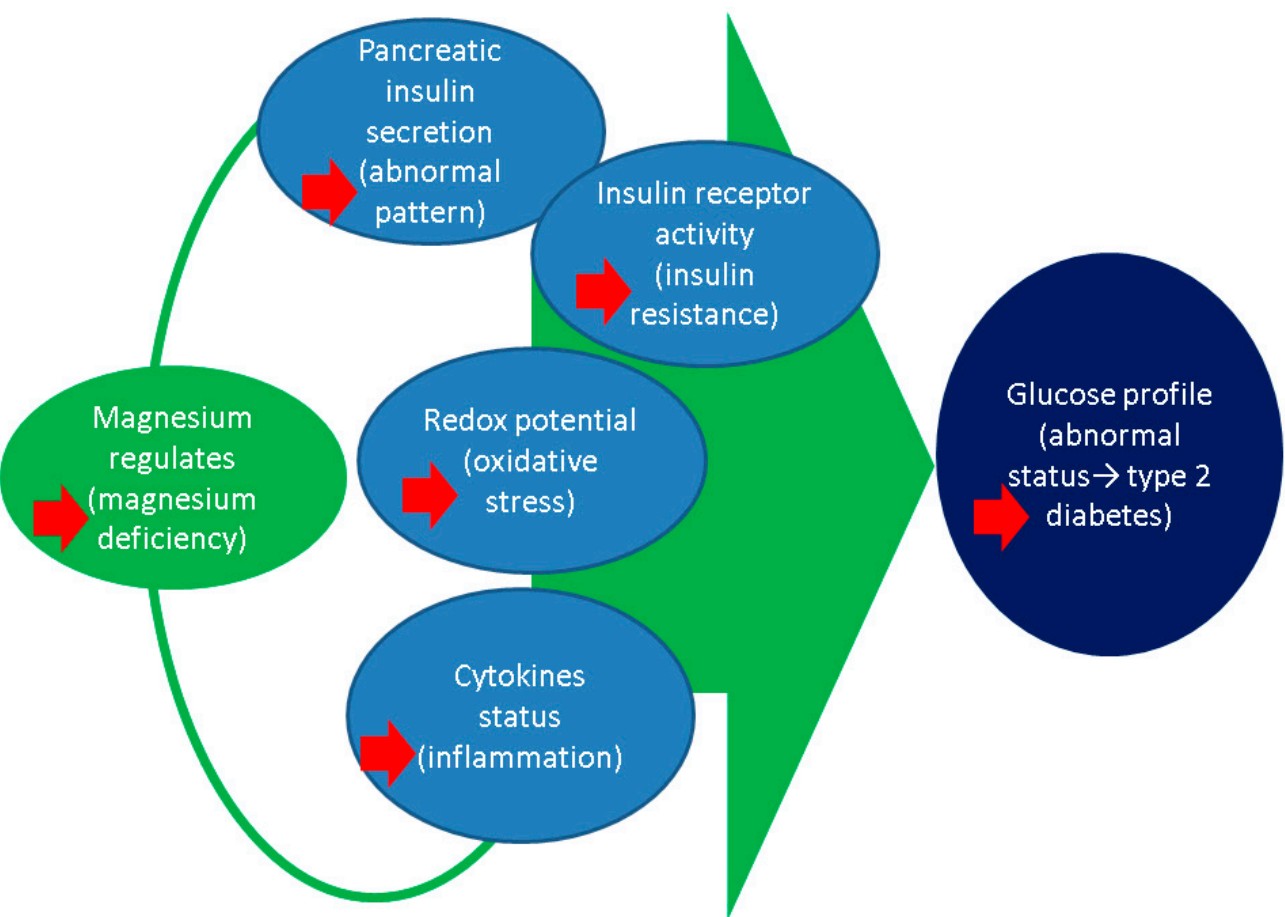

**Figure 1.** Magnesium regulates by being an enzyme co-factor insulin secretion and receptor effects as well as oxidative stress and inflammatory profile, all of which might be impaired amid the detection of type 2 diabetes [22–25].

*Objective*

Our objective was to highlight the most recent clinical data with respect to magnesium intake and metabolic traits, particularly, type 2 diabetes mellitus.

## 2. Methods

This was a PubMed-based narrative review of the English-language medical papers. The key search words included different combinations of terms such as "magnesium" and "intake", "supplementation", respectively, "diabetes", "glucose", "glycaemia", and "insulin resistance". The time frame was from January 2019 until April 2024.

We included (clinically relevant) original studies and excluded cases reports, series, reviews, editorials, opinion, experimental studies in non-type 2 diabetes, and non-human data as well as studies that did not specifically assessed the magnesium intake and only provided assays of serum magnesium, studies on patients diagnosed with type 1 diabetes mellitus or secondary type of diabetes, studies that analyzed metabolic syndrome without providing specific parameters of the diabetic subjects. A total of 30 studies were included and we organized the key findings into three main sections as follows (Figure 2).

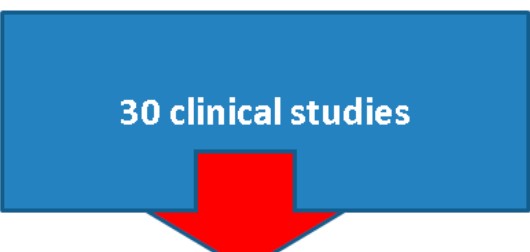

**Figure 2.** The key findings across selected studies according to the mentioned methods.

### 3. Results

*3.1. Magnesium Intake in Diabetic Population*

3.1.1. Adherence to the Recommendations Regarding Daily Magnesium Intake

Studies investigating the magnesium intake in relation to diabetes (n = 2 studies, one transversal and another retrospective cohort, enrolling N = 2823 patients, and 311 of them were individuals with type 2 diabetes) found that most subjects had lower daily nutrient intake than local recommendations [26,27].

Specifically, data regarding type 2 diabetes mellitus were provided by Massimino et al. [26] in a transversal study on 138 subjects diagnosed with this condition (average age of 72 ± 4 years, mean body mass index of 29 ± 5 kg/sqm, approximately 70% were males). Most individuals did not meet the Italian and European recommendations with regard to the nutrient intake per day and had daily means of 160 ± 55 mg in men and 160 ± 49 mg in females, which are below the recommended intakes of 70 mg/day (men and women) and 350 mg/day (men) and 300 mg/day (women), respectively, in the mentioned European countries [26]. Moreover, outside Europe, we identified one study on 2685 subjects that did not find a statistically significant association between the adherence to Taiwan recommendations regarding daily magnesium intake and the prevalence of type 2 diabetes; this was the NAHSIT (Nutrition and Health Survey in Taiwan) cohort from 2013 to 2016 in adults older than 19 years old [27]. The diabetic sub-cohort (N = 173; 57.23% men and 42.77% women) included individuals who were newly diagnosed with type 2 diabetes and the odds ratio between the adherence to dietary reference intake and the risk of having diabetes (that was defined as a glycated haemoglobin A1c (HbA1c) ≥ 6.5% in patients who were not prior known with diabetic disease) was 0.7 [95% CI (confidence interval) between 0.44 and 1.11] in males and 0.89 (95% CI between 0.52 and 1.53) in women, both not being statistically significant (Table 1).

**Table 1.** Data regarding adherence to daily magnesium intake recommendations in subjects with diabetes mellitus [26,27].

| First Author Year of Publication Reference Number | Study Design | Studied Population | Main Findings |
|---|---|---|---|
| Massimino 2023 [26] | Cross-sectional | **N = 138 patients with T2DM** (65.95% men + 34.06% women) | **Mean Mg intake (mg/day)** Men: 160 ± 55 Women: 160 ± 49 **Adherence to Mg intake recommendations**: • **Italian recommendations \*:** Men: 27.5% Women: 36.2% • **European recommendations \*\*:** Men: 0% Women: 2.1% |
| Li 2022 [27] | Retrospective | **N = 2685 patients** (48.2% men + 51.8% women) included: **N1 = 173 patients with T2DM** (57.23% men + 42.77% women): **N2 = 2512 patients without T2DM** (47.53% men + 52.47% women) | **N1:** No association between the **adherence to DRI (Mg) and newly diagnosed T2DM**: Men: OR (95% CI) = 0.70 (0.44–1.11) Women: OR (95% CI) = 0.89 (0.52–1.52) |

Abbreviations: CI = confidence interval; DRI = dietary reference intake; Mg = magnesium; N = number of patients; OR = odds ratio; T2DM = type 2 diabetes mellitus; * 170 mg/day for men and women; ** 350 mg/day for men and 300 mg/day for women; red font = studied groups enrolling patients diagnosed with diabetes mellitus; blue font = specific data with concern to magnesium status/findings.

### 3.1.2. Magnesium Intake and the Risk of Diabetes Mellitus

Deficient dietary nutrient intake was found to be correlated with the risk of developing/having diabetes across five studies [28–32]. The study including the largest number of individuals with diabetes, performed by Han et al. [28], found the lowest prevalence and odds ratio of diabetes in the highest [odds ratio of was 0.56 (95% CI between 0.39 and 0.81); $p < 0.001$] versus the lowest magnesium intake quintile (455.25 versus 156.5 mg/day). This was a National Health and Nutrition Examination Surveys (NHANES) cohort from 2007 until 2018. The studied population included 24,171 US individuals (of 20 years or older) and 9950 of them were diabetic. The diabetes-related analysis was performed using five quintiles according to the daily micronutrient dietary intake, which was negatively correlated not only with the prevalence of diabetes but also hyperlipemia and high blood pressure [28]. Another large cross-sectional study enrolled 20,480 subjects, including 3432 subjects with diabetes mellitus with an average age of 59.4 years and a mean body mass index of 33.1 kg/sqm. The study was conducted by Shah et al. [29], who reported that the highest daily magnesium intake according to the fifth quintile (445 mg/day) was associated with a lower incidence of type 2 diabetes mellitus versus the lowest quintile (152 mg/day) in the overall population ($p < 0.001$), as well as the males subgroup ($p = 0.01$) and the females subgroup ($p < 0.001$), but the association was no longer statistically significant after adjustment for multiple variables including body mass index, education level, physical activity, smoking and drinking status, income, and vitamin D and calcium levels [29]. McClure et al. [30], however, found no statistically significant difference regarding daily nutrient intake between subjects with diabetes mellitus and those without the condition, but did find differences between controlled and uncontrolled diabetic patients [30].

Additional parameters in the magnesium intake–diabetes analysis are represented by the body mass index as well as the mineral metabolism status, particularly calcium and vitamin D profile. For instance, one prospective cohort in 2188 diabetic subjects (an average age of 40 ± 13.3 years and a mean body mass index of 27 ± 4.7 kg/sqm) found different risks of type 2 diabetes mellitus in relation to the daily magnesium intake in distinct sub-groups based on the body mass index; a reduced risk was at an intake of

≥422 mg/day only in subjects with normal weight (hazard ratio of 0.33; 95% CI between 0.14 and 0.84), and no association was confirmed in the overall cohort [31]. A higher risk of type 2 diabetes mellitus in the overall population was also associated with the highest calcium-to-magnesium intake ratio of 4.98 [29]. An interaction with vitamin D was reported in a large cross-sectional study on 10,249 subjects that found an association between vitamin D and a lower incidence of type 2 diabetes mellitus in patients with a higher magnesium intake of more than 267 mg/day. In the lower magnesium intake group, however, there was no statistically significant association between the vitamin D profile and the risk of diabetes mellitus, suggesting that daily magnesium intake plays a role in this interaction, noting that the nutrient represents an essential contributor in the vitamin D activation [32].

Overall, most studies found an inverse association between daily magnesium intake and diabetes mellitus prevalence, but these data are presented amid a rather heterogeneous spectrum that included five studies, one prospective and four of cross-sectional design, enrolling a total of 47,166 studied individuals (Table 2).

**Table 2.** The association between the daily magnesium intake and the risk of diabetes mellitus [28–32].

| First Author Year of Publication Reference Number | Study Design | Studied Population | Daily Mg Intake | Main Findings |
|---|---|---|---|---|
| Han 2024 [28] | Cross-sectional | **N = 9950 patients with DM** | Five quintiles based on daily Mg intake (Q1→Q5) Median (min, max) mg/day Q1: 156.5 (9.5–191) Q5: 455.25 (382–1721) | Adjusted models: **OR (95% CI) between daily Mg intake and DM**: Q2: 0.70 (0.53–0.92) Q3: 0.71 (0.53–0.97) Q4: 0.62 (0.45–0.87) Q5: 0.56 (0.39–0.81) $p < 0.001$ |
| Golmohamadi 2023 [31] | Prospective | **N = 2188 patients with DM** | Five quintiles based on daily Mg intake (Q1→Q5) Median (IQR) = 422 (335–524) mg | **HR (95% CI) daily Mg intake and T2DM**: Q2: 1.09 (0.81–1.49) Q3: 0.76 (0.53–1.3) $p = 0.051$ **HR (95% CI) Mg intake ≥422 mg/day and T2DM according to BMI**: normal weight: 0.33 (0.14–0.84) overweight: 1.02 (0.67–1.53) obese: 1.09 (0.76–1.57) $p = 0.030$ |
| Huang 2021 [32] | Cross-sectional | **N = 10,249 (47.7% men + 52.3% women)** included: **N1 = 5094 patients with Mg intake < 267 mg/day** (26% men + 64% women) **N2 = 5155 patients with Mg intake >267 mg/day** (57.7% men + 42.3% women) | Mean daily Mg intake (mg/day): N: 309.6 ± 3.1 N1: 191.7 ± 12 N2: 411.5 ± 2.5 | **OR (95% CI) vitamin D and DM**: **N1**: 0.925 (0.883–0.970), $p = 0.002$ **N2**: 0.968 (0.919–1.020), $p = 0.225$ $p$ for interaction = 0.001 |

**Table 2.** *Cont.*

| First Author Year of Publication Reference Number | Study Design | Studied Population | Daily Mg Intake | Main Findings |
|---|---|---|---|---|
| Shah 2021 [29] | Cross-sectional | **N = 20,480 patients** included: **N1 = 3432 patients with T2DM** (51.6% me + 48.4% women) **N2 = 17,048 patients without** T2DM (47.4% men + 52.6% women) | Median Mg mg/day according to quintiles: Q1: 152 Q2: 215 Q3: 265 Q4: 328 Q5: 445 | In the overall population: **OR (95% CI) daily Mg intake and T2DM**: Adjustment for race, sex, age: Q2: 0.8 (0.7, 0.9) Q3: 0.8 (0.6, 0.9) Q4: 0.7 (0.5, 0.8) Q5: 0.5 (0.4, 0.6) $p = 0.000$ No statistically significant difference after further multivariable adjustments ($p = 0.699$ and $p = 0.906$ for model 2 and 3) Men: **OR (95% CI) daily Mg intake and T2DM**: Adjustment for race, sex, age: Q2: 0.9 (0.6, 1.2) Q3: 0.8 (0.6, 1.1) Q4: 0.7 (0.5, 0.9) Q5: 0.6 (0.4, 0.8) $p = 0.01$ No statistically significant difference after further multivariable adjustments ($p = 0.566$ and $p = 0.787$ for model 2 and 3) Women: **OR (95% CI) daily Mg intake and T2DM**: Adjustment for race, sex, age: Q2: 0.7 (0.6, 0.9) Q3: 0.7 (0.6, 0.9) Q4: 0.6 (0.5, 0.8) Q5: 0.4 (0.3, 0.6) $p = 0.00$ No statistically significant difference after further multivariable adjustments ($p = 0.936$ and $p = 0.972$ for model 2 and 3) **OR (95% CI) Ca:Mg ratio and T2DM**: Q2: 1.0 (0.8, 1.1) Q3: 1.0 (0.8, 1.2) Q4: 1.1 (0.9, 1.3) Q5: 1.2 (1.0, 1.5) $p = 0.010$ |

**Table 2.** *Cont.*

| First Author Year of Publication Reference Number | Study Design | Studied Population | Daily Mg Intake | Main Findings |
|---|---|---|---|---|
| McClure 2020 [30] | Cross-sectional | **N = 4299 patients** included: **N1 = 947 patients with DM** (51.7% men + 48.3% women) N2 = 3352 patients without DM (48.1% men + 51.9% women) | Mean (95% CI) Mg mg/2000 kcal: N1: 296 (285, 308) N2: 294 (286, 302) *p* = 0.739 | **Daily Mg intake (mean; 95% CI) in mg/2000 kcal** N1 (HbA1c > 9%): 272 (246, 299) N1 (HbA1c < 6.5%): 283 (268, 298) *p* = 0.007 |

Abbreviations: BMI = body mass index; CI = confidence interval; Ca:Mg = calcium/magnesium ratio; DM = diabetes mellitus; HR = hazard ratio; HbA1c = glycated haemoglobin; IQR = interval interquartile; Mg = magnesium; min = minimum; max = maximum; N = number of patients; OR = odds ratio; Q = quintile; red font = studied groups enrolling patients diagnosed with diabetes mellitus; blue font = specific data with concern to magnesium status/findings.

3.1.3. Magnesium Intake and Diabetes Control

Regarding HbA1c levels, the mentioned cohort published by McClure et al. [30] in 2020 found that subjects with HbA1c higher than 9% had 4% lower daily magnesium intake compared to the individuals with levels below 6.5% [30]. Moreover, in subjects with diabetes mellitus, an increased magnesium intake of at least 427 mg/day had a negative linear association with HbA1c index. This was part of the results amid a study from 2024 that included 4249 subjects, more than half being males, with an average age of 45.4 ± 19.5 years, a mean body mass index of 27.8 ± 6.3 kg/sqm, and 11.53% of them being diabetic [33]. Magnesium intake per day may also be involved in complex interactions that impact overall glycaemic control, as suggested by a causal path indicating that higher physical activity level decreased HbA1c through daily magnesium intake, a result that was found in a study on 2831 subjects of various ethnic backgrounds, each of them being diagnosed with diabetes mellitus [34].

On the contrary, we mention two small-sample-size studies (N = 119, respectively, 80) that did not confirm the association between daily magnesium intake and diabetes control [35,36]. Specifically, one of them was a cohort of 119 diabetic subjects (21.85% males and 78.15% females; average age of 54.7 ± 8.4 years; mean body mass index of 34.7 ± 5.5 kg/sqm) that also showed a low daily magnesium intake of less than 67% of the recommended dietary allowance (RDA) in 23.5% of them. No correlation was established between daily magnesium intake and glucose-profile-related parameters such as HbA1c (*p* = 0.249), fasting plasma glucose (*p* = 0.55), and postprandial plasma glucose (*p* = 0.186), yet this dietary parameter was positively associated with lean body mass (r = 0.268, *p* = 0.003) and waist-to-hip ratio (r = 0.213, *p* = 0.02) and displayed borderline significance with regard to a negative correlation with body mass index (r = −0.055, *p* = 0.55) [35]. Moreover, another cohort of 80 females showed, according to the diabetic subgroup (which included 40 subjects with a mean age of 50.1 ± 1.08 years, and a mean body mass index of 32.3 ± 4.92 kg/sqm) versus the non-diabetic sub-group (40 controls with an average age of 49.9 ± 1.07 years, and mean body mass index of 32 ± 5.01 kg/sqm), no correlation between daily magnesium intake and HbA1c levels (r = −0.273, *p* = 0.089) or insulin levels (r = −0.220, *p* = 0.172) [36].

To summarize, four novel studies (N = 7279) in addition to a previously mentioned cohort [30] analysed the relationship between daily nutrient intake and the level of disease control according to various methods (HbA1c, fasting and postprandial glycaemia, and insulin); the association may be linear in diabetic subjects only at certain levels of daily dietary doses. Additionally, the multifactorial influence on HBA1c should take into consideration this dietary determinant, as well, but there are no homogenous and concluding data to confirm its exact influence on HbA1c values (Table 3) [33–36].

**Table 3.** The relationship between daily magnesium intake and diabetes mellitus control [33–36].

| First Author Reference Number Year of Publication | Study Design | Studied Population | Daily Mg Intake | Main Findings |
|---|---|---|---|---|
| Chen 2024 [33] | Cross-sectional | **N = 4249 patients** (51.5% males + 48.7% women) included: **N1** = 11.53% of the N (patients with DM) | **Mg intake and haemoglobin glycation index**: beta (95% CI) = −0.00016 (−0.0003, −0.00003), *p* = 0.019 **N1**: linear dose–response relationship, at ≥427 mg/day (for non-linearity *p* = 0.156) **Non-N1**: L-shaped dose–response relationship, with plateau at ≥495 mg/day (for non-linearity *p* = 0.565) | **Mg intake mg/day**: Median (IQR): overall: 248 (176–345) men: 284 (200–389.5) women: 212 (156–292) *p* < 0.001 |
| Kocyigit 2023 [36] | Case–control | **N = 80 females** included: **N1 = 40 patients with T2DM** N2 = 40 controls | Mean ± SD mg/day **N1**: 376.4 ± 103.62 **N2**: 402.2 ± 117.97 *p* = 0.303 **% of dietary reference index** **N1**: 67.5% **N2**: 75% *p* = 0.459 | **N1**: no correlation between **daily Mg intake** and: **HbA1c**: r = −0.273, *p* = 0.089 **Insulin**: r = −0.220, *p* = 0.172 |
| Wu 2020 [34] | Retrospective | **N = 2831 patients with T2DM** on oral medication included **Non-Hispanic White: N1** = 1572 (46.7% men + 53.3% women) **Non-Hispanic Black: N2** = 814 (44.2% men + 55.8% women) **Mexican American: N3** = 445 (51.5% men + 48.5% women) | **N1**: Males: mean = 326.1 ± 168.7 mg/day N1: Females: mean = 265 ± 125 mg/day *p* < 0.001 **N2**: Males: mean = 288.7 ± 159.5 mg/day N2: Females: mean = 247.4 ± 128.9 mg/day *p* < 0.001 **N3**: Males: mean = 367.1 ± 196.7 mg/day N3: Females: mean = 293.4 ± 131.8 mg/day *p* < 0.001 | Path coefficients: Physical Activity > Mg: 0.205 Physical Activity > Mg > HbA1c: −0.026 Education > Mg: 0.152 Education > Mg > HbA1c: −0.019 Gender > Mg: 0.373 Gender > Mg > HbA1c: −0.047 |
| Ozcaliskan 2019 [35] | Cross-sectional | **N = 119 with T2DM** (21.85% men + 78.15% women) | <67% of the recommended daily allowance in 23.5% of subjects | **Correlation between daily Mg intake and the following**: **Lean body mass**: r = 0.268, *p* = 0.003 **Waist-to-hip ratio**: r = 0.213, *p* = 0.020 **HbA1c**: r = −0.107, *p* = 0.249 **Fasting plasma glucose**: r = −0.056, *p* = 0.550 **Postprandial plasma glucose**: r = −0.136, *p* = 0.186 **BMI**: r = −0.055, *p* = 0.550 **Waist circumference**: r = 0.156, *p* = 0.091 **Body fat percentage**: r = −0.149, *p* = 0.107 |

Abbreviations: BMI = body mass index; DM = diabetes mellitus; HbA1c = glycated haemoglobin; IQR = interval interquartile; Mg = magnesium; N = number of patients; red font = studied groups enrolling patients diagnosed with diabetes mellitus; blue font = specific data with concern to magnesium status/findings.

3.1.4. Analysing Diabetes-Related Complications and Mortality with Regard to Magnesium Intake

Bahrampour et al. [37] studied the relationship between daily magnesium intake and diabetic nephropathy in a case–control study on 210 females (105 patients with this complication; a mean age of $55.53 \pm 7.04$ years and 105 controls with an average age of $55.41 \pm 7.14$ years) and found that the risk of this mentioned complication was lowered by a diet with higher amounts of minerals including magnesium with a mean of $327.35 \pm 33.26$ mg/day. The factor loading of magnesium for this pattern was 0.794 [37]. Diabetic retinopathy and magnesium intake were the object of two studies [38,39]. Both in Xu et al.'s [38] retrospective analysis of 3794 diabetic patients and in Zhang et al.'s [39] cross-sectional study on 4596 diabetic subjects a higher daily magnesium intake reduced the odds ratio of diabetic retinopathy [38,39].

Not only does lower magnesium intake per day increase the prevalence of diabetes mellitus, but it has also been linked to a higher mortality in these patients. Wang et al. [40] found in a retrospective analysis on 2045 diabetic subjects (average age of $52.9 \pm 10.1$ years, a mean body mass index of $33.6 \pm 8$ kg/sqm) that an intake of less than 250 mg/day increased the risk both of all-cause mortality and of mortality of other causes excluding cardiovascular and neoplasia [40]. This connection was also shown in a prospective study on 4699 diabetic individuals whereas higher dietary magnesium intake was associated with lower all-cause mortality [41].

To conclude, we identified three studies related to the daily dietary determinant and diabetic complications (n = 3 studies of different designs, namely, one cross-sectional, one prospective, and another case–control study) in terms of retinopathy (n = 1, N = 3794) and nephropathy (n = 2, N = 4805) that suggested a lower intake was associated with a higher risk of such complications. Additionally, two other studies (one prospective and one retrospective cohort) focused on mortality (N = 6744), which, taking only certain mortality indicators into consideration, might be decreased in the subgroups with a higher micronutrient intake (Table 4) [37–41].

**Table 4.** The relationship between daily magnesium intake and diabetes-related complications and mortality [37–41].

| First Author Reference Number Year of Publication Study Design | Studied Population | Daily Mg Intake | Main Findings |
|---|---|---|---|
| **Diabetes-related complications** | | | |
| Bahrampour 2023 Case–control study [37] | **N = 210 females with T2DM** included: **N1 = 105** patients with diabetic nephropathy **N2 = 105** controls | **High minerals group:** $327.35 \pm 33.26$ mg **Low minerals group:** $291.66 \pm 48.31$ mg **Principal factor loading of Mg for the mineral pattern** = 0.794 | OR (95% CI) high mineral pattern and diabetic nephropathy: Crude model: 0.56 (0.32–0.97), $p = 0.03$ Adjusted model: 0.51 (0.28–0.95), $p = 0.03$ |

**Table 4.** *Cont.*

| First Author Reference Number Year of Publication Study Design | Studied Population | Daily Mg Intake | Main Findings |
|---|---|---|---|
| Xu 2023 Retrospective study [38] | **N = 3794 patients with DM**: **N1 = 791 patients with diabetic retinopathy** (52.1% men + 47.9% women) **N2 = 3003 patients without retinopathy** (52.26% men + 47.44% women) | Median (IQR) **N1**: 250 (190–340) mg/day **N2**: 270 (200–360) mg/day *p* = 0.009 Q1: <200 mg/day Q2: 200–270 mg/day Q3: 270–360 mg/day Q4: ≥360 mg/day | OR (95% CI) Mg intake and diabetic retinopathy, adjusted for multiple variables: Q2: 0.73 (0.51–1.06), *p* = 0.093 Q3: 0.62 (0.43–0.92), *p* = 0.016 Q4: 0.48 (0.32–0.73), *p* = 0.001 U-shaped association between Mg intake and diabetic retinopathy: Stable risk when Mg reaches 380 mg/day (OR = 0.76, 95% CI: 0.60–0.76) Association not statistically significant for Mg over 480 mg/day (OR = 0.72, 95% CI: 0.52–1.00) |
| Zhang 2022 Cross-sectional study [39] | **N = 4595 patients with DM** (49.49% men + 50.51% women) | NA | **Univariate regression**: **OR (95% CI) Mg intake and diabetic retinopathy** Q2: 0.95 (0.77–1.17) Q3: 0.91 (0.73–1.13) Q4: 0.77 (0.61–0.96) Q5: 0.79 (0.64–0.99) **Multivariate regression**: **OR (95% CI) Mg intake and diabetic retinopathy adjusted for sex and age**: Q2: 0.94 (0.76–1.17) Q3: 0.90 (0.72–1.12) Q4: 0.76 (0.60–0.95) Q5: 0.78 (0.62–0.98) |
| | | **Diabetes-related mortality** | |
| Wang 2022 Retrospective study [40] | **N = 2045 patients with DM** (49.1% men + 50.9% women) | ≤250 mg =983 (45.62%) >250 mg =1062 (54.38%) | **HR (95% CI) Mg intake ≤250 mg/day and** **all-cause mortality**: 1.56 (1.13–2.16) *p* < 0.01 **other-cause mortality**: 1.68 (1.09–2.60) *p* < 0.05 |
| Wang 2022 Prospective cohort [41] | **N = 4699 patients with DM** According to dietary antioxidant index (three tertiles) | Mean = 265.70 ± 1.71 mg/day | **HR (96% CI) daily Mg intake and:** **all-cause mortality**: tertile 2: 0.75 (0.61–0.91) tertile 3: 0.65 (0.51–0.81) *p* < 0.001 **cardiovascular mortality:** tertile 2: 0.71 (0.52–0.98) tertile 3: 0.69 (0.41–1.09) *p* = 0.093 **cancer mortality:** tertile 2: 0.59 (0.36–0.96) tertile 3: 0.8 (0.45–1.4) *p* = 0.460 |

Abbreviations: CI = confidence interval; DM = diabetes mellitus; HR = hazard ratio; IQR = interval interquartile; Mg = magnesium; N = number of patients; NA = not available; OR = odds ratio; Q = quintile; red font = studied groups enrolling patients diagnosed with diabetes mellitus; blue font = specific data with concern to magnesium status/findings.

*3.2. Insulin Resistance and Others Assessments of Glucose Profile in the General Population with Respect to Magnesium Intake*

When investigating the connection with daily magnesium intake, most studies used the HOMA-IR (Homeostatic Model Assessment for Insulin Resistance) as an indicator of insulin resistance. In the prior mentioned case–control study on 40 cases with type 2 diabetes mellitus and 40 controls, HOMA-IR was also negatively associated with magnesium intake per day (r = −0.393, $p$ = 0.012) [36]. The lower HOMA-IR in people with higher daily magnesium intake was found not only in diabetic patients but also in normoglycemic and pre-diabetic subjects according to a cross-sectional study from 2022 (N = 10,609 individuals) [42]. Moreover, a large cross-sectional study on 8120 subjects from the Chinese population found a negative association between daily magnesium intake and HOMA-IR, with the lowest risk in the highest magnesium-related quartile. The prevalence ratio was 0.250 [(95% CI between 0.204 and 0.306), $p$ < 0.001] [43]. Bavani et al. [44], however, did found no association between magnesium intake and HOMA-IR, nor with another indicator of insulin resistance, QUICKI (quantitative insulin sensitivity check index), after adjustment for endothelial dysfunction parameters, in a cross-sectional study on 345 females [44]. Moreover, the triglyceride index did not have a statistically significant association with magnesium intake in a cross-sectional study on 778 subjects (mean age of 44.9 ± 10.6 years) [45]. The largest study investigating the relationship between magnesium intake and fasting plasma glucose was performed by Yan et al. [46] on 8322 subjects (51.1% of them were aged between 30 and 59 years and 48.9% were of 60 years old or older) and found negative correlations between daily magnesium intake and fasting plasma glucose, both in men and in women, with the strongest associations in the highest magnesium intake quantiles (r = −5.208) in men and (r = −9.674) in women [46]. A negative correlation between daily magnesium intake and plasma glucose [beta (96% CI) = 0.072 (−0.019, 0.405), $p$ = 0.074] was also found in a cross-sectional study on 2373 adults. Magnesium intake per day did not correlate with HbA1c [47]. Even though Bentil et al. [48] found an association between daily magnesium intake and fasting plasma glucose and HbA1c according to a transversal (pilot) study on 63 females, the statistical significance was not confirmed after further adjustment for age and body mass index [48].

As a short note, seven studies (N = 30,610 individuals) [42–48] and another one that was previously specified [36] analysed the perspective of daily magnesium intake in the general population with the endpoint of different features of the glucose profile, particularly insulin resistance. Concerning HOMA-IR, there were three confirmatory studies [36,42,43] and one non-confirmatory [44], while fasting plasma glucose was highlighted as inversely correlated with daily nutrient intake in two studies [46,47] and not confirmed in another small study (Table 5) [48].

**Table 5.** The relationship between daily magnesium intake and insulin resistance and other glucose profile elements in the general population [36,42–48].

| First Author Reference Number Year of Publication Study Design | Studied Population | Daily Mg Intake | Main Findings |
|---|---|---|---|
| **Insulin resistance in general population** | | | |
| Kocyigit ** 2023 Case–control study [36] | **N = 80** females included: **N1 = 40 patients with T2DM** **N2 = 40 controls** | Mean ± SD mg/day N1: 376.4 ± 103.62 N2: 402.2 ± 117.97 $p$ = 0.303 | **N1:correlation between daily Mg intake and HOMA-IR**: r = −0.393, $p$ = 0.012 |

**Table 5.** *Cont.*

| First Author Reference Number Year of Publication Study Design | Studied Population | Daily Mg Intake | Main Findings |
|---|---|---|---|
| Mirrafiei 2022 Cross-sectional study [45] | **N = 778 individuals** (232 males + 546 females) | Mean ± SD mg/day Males: 294 ± 140 mg/day Females: 262 ± 112 mg/day $p < 0.001$ | No statistically significant association between **daily Mg intake and**: **TyG**: men β ± SE = 0.000 ± 0.000, $p = 0.61$ women β ± SE = −0.000 ± 0.000, $p = 0.55$ |
| Palacios 2022 Case–control study [42] | **N = 10,609 individuals** included: **Prediabetes N1 = 3169** (48.8% men + 51.2% women) **Normoglycemia N2 = 7740** (48.4% men + 51.6% women) | **N1 ***: Q1 < 124.72 mg/day Q2: between 124.72 and 155.68 mg/day Q3: between 155.69 and 193.80 mg/day Q4 > 193.80 mg/day **N2 ***: Q1 < 117.21 mg/day Q2: between 117.21 and 145.18 mg/day Q3 between 145.19 and 182.47 mg/day Q4 > 182.47 mg/day | **Geometric mean (SE) HOMA-IR based on daily Mg intake quartiles**: **N1**: Q1: 1.14 (0.03) Q2: 1.10 (0.03) Q3: 1.02 (0.05) Q4: 0.90 (0.03) $p < 0.0001$ **N2**: Q1: 0.78 (0.03) Q2: 0.79 (0.02) Q3: 0.71 (0.02) Q4: 0.61 (0.03) $p < 0.0001$ |
| Bavani 2021 Cross-sectional study [44] | **N = 345 females** | **Mg intake (mg/kg) according to quintiles**: **Mean, SE (mg/day)** Q1: 205, 7 Q2: 221, 8 Q3: 254, 7 Q4: 355, 9 | **Mean (SE) QUICKI based on daily Mg intake quartiles**: Multivariable adjustment: Q1: 0.34 (0.02) Q2: 0.36 (0.01) Q3: 0.40 (0.01) Q4: 0.39 (0.02) $p = 0.02$ After adjustment for endothelial dysfunction parameters, $p = 0.09$ No statistically significant association between daily Mg intake and fasting plasma glucose ($p = 0.93$), insulin ($p = 0.08$), HOMA-IR ($p = 0.11$) |
| Yang 2020 Cross-sectional study [43] | **N = 8120 individuals** | **Mean Mg intake (mg/kg) according to quintiles**: Q1: 2.91 ± 0.52 Q2: 4.09 ± 0.28 Q3: 5.14 ± 0.36 Q4: 7.66 ± 2.06 Mean (mg/day) Q1: 190.61 ± 46.16 Q2: 254.04 ± 45.52 Q3: 301.93 ± 52.99 Q4: 426.05 ± 124.43 | **PR (95% CI) daily Mg intake per kg body weight and insulin resistance** (adjustment for multiple variables) Q2: 0.589 (0.511, 0.678) Q3: 0.421 (0.360, 0.493) Q4: 0.250 (0.204, 0.306) $p < 0.001$ |

**Table 5.** *Cont.*

| First Author Reference Number Year of Publication Study Design | Studied Population | Daily Mg Intake | Main Findings |
|---|---|---|---|
| **Glucose profile in general population** | | | |
| Xu 2023 Cross-sectional study [47] | **N = 2373 individuals** (37.5% men + 62.5% women) | Median (IQR): 205.94 (144–249) mg/day | Adjusted models for relationship between the following: **Daily Mg intake and HbA1c**: beta (96% CI): overall: 0.072 (−0.019, 0.405), $p = 0.074$ men: 0.132 (−0.007, 0.771), $p = 0.054$ women: 0.033 (−0.165, 0.330), $p = 0.512$ **Daily Mg intake and fasting plasma glucose**: beta (96% CI): overall: −0.087 (−0.512, −0.034), $p = 0.025$ men: −0.061 (−0.610, 0.221), $p = 0.358$ women: −0.098 (−0.597, −0.014), $p = 0.040$ |
| Bentil 2021 Cross-sectional, pilot study [48] | **N = 63 women** Mean age = 29.5 ± 8.5 years BMI: underweight 1.6%, normal weight 54%, overweight 30.2%, obese 14.3% | Mean = 200 ± 116 mg 15.9% met RDA | **Association between daily Mg intake and fasting blood glucose**: Beta coefficient [95% CI]: unadjusted model: 0.31 (0.07–0.55), $p = 0.01$ adjusted for age and BMI: 0.22 (−0.03–0.46), $p = 0.08$ **Association between daily Mg intake and HbA1c**: Beta coefficient [95% CI]: unadjusted model: 0.26 (0.01–0.51), $p = 0.04$ adjusted for age and BMI: 0.15 (−0.08–0.39), $p = 0.2$ |
| Yan 2019 Cross-sectional study [46] | **N = 8322 individuals** (50.1% men + 49.9% women) | Mean (95% CI) Males: 372.71 (362.48, 382.94) mg/day Females: 305.34 (294.20, 316.48) mg/day | **Regression coefficients for daily Mg intake and fasting plasma glucose quantiles**: Men: 0.1: −4.165 (−5.612, −0.411) 0.2: −4.926 (−8.654, −1.038) 0.3: −5.093 (−8.407, −1.436) 0.4: −4.670 (−7.303, −0.611) 0.5: −3.387 (−6.739, −0.480) 0.6: −1.803 (−7.486, −0.182) 0.7: −3.473 (−7.141, −1.137) 0.8: −4.036 (−6.633, −2.231) 0.9: −5.208 (−11.844, −2.033) $p < 0.05$ Women: 0.3: −4.300 (−7.034, −0.739) 0.4: −3.244 (−6.630, −0.913) 0.6: −2.141 (−5.627, −0.214) 0.7: −3.983 (−8.388, −0.616) 0.8: −7.295 (−11.42, −1.695) 0.9: −9.674 (−16.319, −0.700) $p < 0.05$ |

Abbreviations: BMI = body mass index; CI = confidence interval; DM = diabetes mellitus; HR = hazard ratio; HbA1c = glycated haemoglobin; HOMA-IR = Homeostatic Model Assessment for Insulin Resistance; IQR = interval interquartile; Mg = magnesium; N = number of patients; PR = prevalence ratio; RDA = recommended dietary allowance Q = quintile; SD = standard deviation; SE = standard error; TyG = triglyceride glucose index; ** this is the same study as prior table; *** values standardized for 1000 kcal; red font = studied groups enrolling patients diagnosed with diabetes mellitus; blue font = specific data with concern to magnesium status/findings.

### 3.3. The Effects of Magnesium Supplementation on Glycaemic Traits

The highest level of evidence regarding magnesium-supplementation-related effects on glucose metabolism and metabolic syndrome comes from seven randomized controlled trials (N = 350 patients). However, the sample size was reduced (from 14 to 86 individuals per study, either diabetic or pre-diabetic) and the outcomes were rather discordant [49–55]. Further analysis of long-term outcomes in the field of magnesium supplementation is mandatory. On one hand, both fasting plasma glucose and insulin decreased in patients with diabetes mellitus who received supplementation with 250 mg of magnesium oxide and 150 mg of zinc sulphate, as revealed by a double-blind randomized controlled trial performed by Hamedifard et al. [49]. HOMA-IR improved by supplementation with 350 mg of magnesium daily from 440 mL of balanced deep-sea water in a randomized double-blind crossover trial performed by Ham et al. [50] on 37 subjects with prediabetes. Fasting plasma glucose, however, did not have a statistically significant change [50]. A lowering in HbA1c levels in people with metabolic syndrome ($-0.28 \pm 0.27$, $p = 0.0036$) was also reported following supplementation with 400 mg magnesium daily [51]. On the other hand, other studies did not find a better glycaemic control or improved HOMA-IR. For instance, in Drenthen et al.'s [52] study on 14 subjects with type 2 diabetes and insulin treatment, daily supplementation with 360 mg of magnesium from 150 mL magnesium gluconate solution did not modify insulin dose ($p = 0.869$) or HbA1c ($p = 0.851$) compared to placebo [52]. In subjects with prediabetes, in a study performed by Salehidoost et al. [53], HOMA-IR, HbA1c, and fasting plasma glucose did not have statistically significant differences between placebo and supplementation with 250 mg magnesium oxide [53]. Neither HOMA-IR nor QUICKI were improved by supplementation with 250 mg Mg oxide and 150 mg zinc sulphate [49]. Two studies reported data regarding diabetic nephropathy and magnesium supplementation [54,55]. A double-blind randomized controlled trial on 80 subjects with diabetes mellitus and early-stage diabetic nephropathy showed that 250 mg of magnesium oxide reduced microalbuminuria, but the change was not statistically significant ($p = 0.09$). However, supplementation with magnesium increased insulin resistance, but did not alter glycaemic control as measured by HbA1c and fasting plasma glucose [54]. Another study found improved urinary-albumin-to-creatinine ratio, as well as improved glomerular filtration rate, in subjects with diabetic nephropathy who received supplementation with 360 mg magnesium from 2.25 g magnesium citrate (Table 6) [55].

**Table 6.** Interventional studies: magnesium supplementation and glucose profile effects [49–55].

| First Author Reference Number Year of Publication Study Design | Studied Population | Mg Supplementation Dose Per Day | Outcome |
|---|---|---|---|
| Drenthen 2024 Randomized controlled trial [52] | **N = 14 patients (50% women) with insulin-treated T2DM and hypomagnesaemia** (serum Mg $\leq 0.79$ mmol/L) Mean age = 67 ± 6 years Mean BMI = 31 ± 5 kg/m$^2$ | Supplementation with 150 mL Mg gluconate (=360 mg Mg) | **Mg versus placebo**: **insulin dose** $p = 0.869$ **HbA1c** $p = 0.851$ |
| Salehidoost 2022 Randomized controlled trial [53] | **N = 86 patients with prediabetes** (21.1% men + 78.9% women) Magnesium supplementation: **N1 = 37** (24.33% men + 75.67% women) Mean age = 56.7 ± 5.9 years Placebo: **N2 = 34** (17.65% men + 82.35% women) Mean age = 54.8 ± 4.9 years | Supplementation with 250 mg Mg oxide | **Mg versus placebo**: **HOMA-IR** $p = 0.17$ **HbA1c** $p = 0.63$ **fasting plasma glucose** $p = 0.57$ |

**Table 6.** *Cont.*

| First Author Reference Number Year of Publication Study Design | Studied Population | Mg Supplementation Dose Per Day | Outcome |
|---|---|---|---|
| Ham 2020 Randomized double-blind and crossover trial [50] | **N = 37 patients with prediabetes** | Supplementation with 350 mg Mg | **Balanced deep-sea water with 350 mg Mg/440 mL versus placebo**: **change in HOMA-IR**: $-0.27 \pm 1.01$ versus $0.17 \pm 0.70$, $p = 0.049$ **change in fasting plasma glucose**: $0.73 \pm 4.88$ versus $0.46 \pm 6.91$, $p = 0.837$ |
| Hamedifard 2020 Randomized double-blind controlled trial [49] | **N = 55 females with T2DM and coronary heart disease** Mg and zinc: N1 = 27 Mean age = $61.7 \pm 9.4$ years Placebo: N2 = 28 Mean age = $62.6 \pm 10.8$ years | Supplementation with 250 mg Mg oxide + 150 mg Zn sulphate | **Supplementation with 250 mg Mg oxide and 150 mg Zn sulphate versus placebo** β (95% CI): **fasting plasma glucose:** $-9.44$ ($-18.30$, $-0.57$), $p = 0.03$ **insulin:** $-1.37$ ($-2.57$, $-0.18$), $p = 0.02$ HOMA-IR: $-0.36$ ($-0.75$, $0.02$), $p = 0.06$ **QUICKI:** $0.006$ ($-0.002$, $0.01$), $p = 0.12$ |
| Afitska 2021 Randomized controlled trial [51] | **N = 24 patients with metabolic syndrome and normal serum Mg** (41.7% men + 58.3% women) **N1 = 13** (39% men + 61% women) Mean age = $61.8 \pm 10.7$ years **Placebo**: N2 = 11 (45% men + 55% women) Mean age = $71.9 \pm 7.8$ years | Supplementation with 400 mg Mg citrate | **Change in HbA1c in Mg group**: $-0.28 \pm 0.27$, $p = 0.0036$ **Mg versus placebo**: $p = 0.02$ |
| **Specific outcomes in diabetic nephropathy** | | | |
| Halawa 2023 Randomized controlled trial [55] | **N = 54 patients with T2DM and nephropathy** (59% men + 41% women) **(Mg) N1** = 26 (61.5% men + 38.5% women) Mean age = $61.4 \pm 7.5$ years (**Control**) N2 = 28 (57.1% men + 42.9% women) Mean age = $63 \pm 7.2$ years | Supplementation with 2.25 g Mg citrate (=360 mg Mg) | **Mg supplementation versus placebo** → percent change in: **urinary-albumin-to-creatinine ratio:** $-6.87$ ($-9.17$, $-4.84$) versus $-0.9$ ($-1.797$, $-1.58$), $p = 0.001$ **eGFR**: $21.74$ ($12.14$, $37.41$) versus $0$ ($-4.01$, $-0$), $p = 0.001$ |
| Sadeghian 2020 Randomized double-blind, controlled trial [54] | **N = 80 patients with T2DM and early-stage diabetic nephropathy** **(Mg) N1 = 40** (65.9% women, 34.1% men) Mean age = $41.2 \pm 8.8$ years (**Placebo**) **N2 = 40** (67.5% women, 32.5% men) Mean age = $42.8 \pm 8.4$ years | Mg supplementation group: mean dietary intake = $195.5 \pm 77.7$ mg/day Placebo group: mean Mg dietary intake = $210.5 \pm 160.3$ mg/day | **Mg supplementation with 250 mg Mg oxide versus placebo** → change in: **microalbuminuria**: $-14 \pm 9.9$ versus $-3.1 \pm 2.2$, $p = 0.09$ **HbA1c**: $0.09 \pm 1.3$ versus $0.29 \pm 0.1$, $p = 0.36$ **fasting plasma glucose**: $-1.4 \pm 58.5$ versus $-7.9 \pm 39.2$, $p = 0.47$ **HOMA-IR**: $1.9 \pm 4$ versus $0.2 \pm 2.2$, $p = 0.04$ |

Abbreviations: BMI = body mass index; DM = diabetes mellitus; eGFR = estimated glomerular filtration rate; HbA1c = glycated haemoglobin; HOMA-IR = Homeostatic Model Assessment for Insulin Resistance; Mg = magnesium; N = number of patients; RDA = recommended dietary allowance; QUICKI (quantitative insulin sensitivity check index); Zn = zinc; red font = studied groups enrolling patients diagnosed with diabetes mellitus; blue font = specific data with concern to magnesium status/findings.

## 4. Discussion

### 4.1. Magnesium Intake and Metabolic Features: Beyond Diabetes Mellitus

Nutrient intake has been found to correlate not only with glucose profile, but also with other cardio-metabolic features, thus the importance of addressing this topic and its complexity. Particularly, the issue of type 2 diabetes mellitus cannot be regarded as an isolated ailment and the entire metabolic and cardiovascular panel should be analysed in one patient or one population, including in relation with the magnesium status interplay, specifically, nutrient intake and deficiency. This should help clinicians to have a complex multidisciplinary approach and to assess clinical management through an in-depth perspective, which can only help patients to achieve an overall better outcome. For this, we looked at the tidily connected data on metabolic syndrome and cardiovascular risk with regard to magnesium intake (Figure 3).

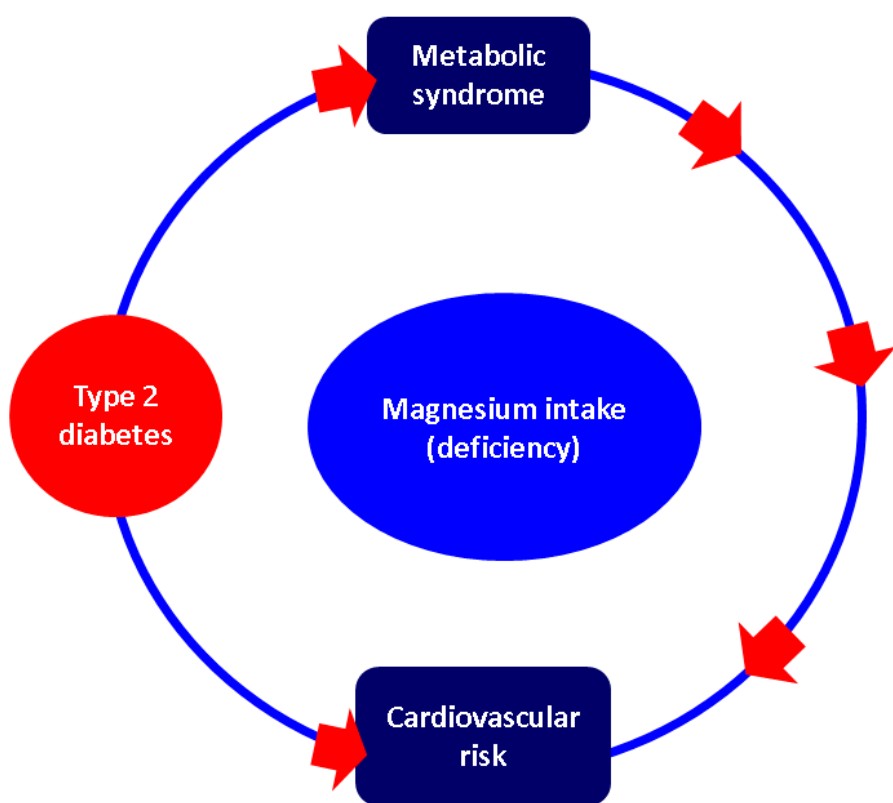

**Figure 3.** Sneak peak of magnesium interplay and type 2 diabetes mellitus amid the co-presence of metabolic syndrome and associated high cardiovascular risk.

### 4.1.1. Delving into Obesity and Metabolic Syndrome

Higher daily magnesium intake decreased the prevalence of metabolic syndrome in numerous studies. For instance, a negative association between dietary nutrient intake and metabolic syndrome was reported in a large longitudinal study by Dibaba et al. [56] in 2021 (N = 6802 subjects across REGARDS cohort) [56]. A lower prevalence of metabolic syndrome in subjects with elevated magnesium intake was also confirmed by Yang et al. [43] in the Chinese population (the glycaemic parameters across this cohort were already mentioned) [43]. A large prospective study on 1114 normal-weight and over-weight/obese adults showed that a higher dietary concentration was associated with a reduction in the risk of developing more than one metabolic syndrome component [57]. Moreover, logistic regression analysis in 124 metabolically healthy obese (having a maximum of one component of the metabolic syndrome criteria) versus 124 metabolically unhealthy obese (adjusted for sex and age) revealed a statistically significant correlation between the first type of metabolic profile and magnesium intake. The odds ratio was 1.17, with a 95% CI

between 1.07 and 1.25 [58]. In a longitudinal study published in 2022, Jiao et al. [59] found that the highest magnesium intake (of ≥329.91 mg/day) decreased the risk of metabolic syndrome. Moreover, this cohort identified a U-shaped relationship with a lower risk of metabolic syndrome at an intake of magnesium lower than 280 mg/day, suggesting that the dose–effect relationship of magnesium intake on metabolic syndrome may be more complex and it depends on specific cut offs [59]. This might explain why some authors revealed on the contrary a negative impact on metabolic complications and these results potentially relate to a certain daily dietary range. For instance, Zhu et al. [60] found that magnesium intake was positively associated with metabolic syndrome in a cross-sectional study on 5323 subjects. This association, in opposition to prior findings, was attributed by the authors to the high amounts of magnesium coming from grain and potato consumption as a possible bias [60].

Independently of the entire panel of metabolic syndrome criteria, an insufficient magnesium intake was also found in over-weight and obese subjects compared with normal-weight subjects in a cross-sectional study from 2020 (N = 2969 adults), an analysis coming from NHANES, between 2011 and 2014 [61]. Similar results were reported by Lin et al. [62], but only in women with obesity and not in men (a nationwide cohort in Taiwan). In females, adherence to daily magnesium intake recommendations was positively associated with body mass index. The odds ratio was 0.72; 95% CI was between 0.54 and 0.95 for a body mass index of ≥27 kg/sqm. It was also associated with waist circumference (odds ratio of 0.78; 95% CI between 0.64 and 0.97, $p < 0.05$) [62]. The largest study providing data regarding the link with obesity was a cross-sectional cohort conducted by Jiang et al. [63] on 19,952 subjects, with a median body mass index of 27.9 kg/sqm; the authors found that both body mass index and waist circumference were inversely associated with dietary magnesium intake per day [63]. Another large prospective study on 4097 subjects (young American adults) found the lowest hazard ratio of incident obesity in the highest magnesium quintile (Q5 = 193.6–238.2 mg/1000 kcal) as well as an inverse association between daily magnesium intake and body mass index after 30 years [64].

Studies showed that lower magnesium intake was associated with obesity, as defined by not only calculating the body mass index but also by assessing the body fat. Elevated magnesium intake might lead to a lower visceral fat accumulation [65]. The association between low micronutrient intake and high body fat was also found in subjects with impaired glucose tolerance [66]. Of note, similar correlations have been established with other obesity indices such as body adiposity, abdominal volume, body roundness, and weight-adjusted waist, as well as coronary indices [67].

In contrast with most of the findings, Mehta et al. [68] identified a higher magnesium intake in males with obesity (mean daily intake of 254.9 ± 101.1 mg/day) compared with normal-weight men (mean intake of 242 ± 102.2 mg/day). In women, however, the difference was not statistically significant [68]. Similarly, Liu et al. [69] did not confirm any differences in terms of daily magnesium intake according to the obesity status, neither in Non-Hispanic Black people nor in Mexican American people aged between 19 and 39 years (this was a NHANES sub-analysis, between 2003 and 2014) [69]. The relationship between the intake of this nutrient and obesity might be in fact multifactorial. Hence, we mention some data on obese adults, whereas a negative correlation between magnesium intake per day and serum parathyroid hormone and 25-hydroxyvitamin D levels, as well as interleukin 6, was confirmed (r = −0.295, $p < 0.05$) [70].

To summarize, a lower magnesium intake was linked to an elevated prevalence of diabetes mellitus, obesity, and metabolic syndrome in most studies. The adherence to daily recommendations was decreased in diabetic people with or without obesity. However, a large variability in assessing the adherence to optimum nutrient doses was noted, and this might come as a potential bias. As highlighted, a magnesium intake of more than 400 mg/day was associated with a lower prevalence of diabetes mellitus than a dose of 455.25 mg/day [28], respectively, of 445 mg/day [29], while a reduced HOMA-IR was

identified at a value of above 402.2 mg/day [36], and a lower prevalence of metabolic syndrome at a dose of more than 437.90 mg/day [56].

Whether specific tailoring of daily magnesium intake recommendations to subgroups in terms of age and sex should be differently approached with regard to each metabolic component is still an open issue. From the clinical to molecular levels, we should specify that the relationship between magnesium and insulin secretion as well as insulin resistance is intricate and still incompletely understood [71,72]. On the one hand, magnesium is involved in pathways leading to increased insulin secretion. Due to its role in regulating glucokinase, it is involved in the metabolism of glucose-6-phosphate, increasing ATP (adenosine triphosphate) production, leading to the closure of ATP-sensitive potassium channels, which generates calcium influx, and promoting the release of insulin granules. Magnesium is also involved in the activation of protein kinase C that activates adenylate cyclase, leading to an increase in cAMP (cyclic adenosine monophosphate) that binds and activates protein kinase A, which in turn increases the release of insulin granules [71–74]. On the other hand, magnesium and the magnesium–ATP complex keep the ATP-sensitive potassium channels open. Moreover, magnesium inhibits the calcium channels. These effects lower calcium influx, and therefore lower insulin release [71]. Magnesium may also play a role in the preservation of ß-cell insulin secretion, as suggested by the influence of extracellular magnesium on NIPAL1 (Nuclear Interaction Partner of Alkaline Phosphatase-like domain containing 1) expression. Lower cellular magnesium was linked to the downregulation of the NIPAL1 transporter which in turn was associated with a reduced insulin secretion and intracellular storage [75].

Additionally, magnesium is also involved in the binding of insulin to its receptor and the insulin signal transduction pathways [76]. Overall, the involvement of magnesium in mechanisms leading to the inhibition of insulin secretion, the impairment of insulin signalling, chronic inflammation, and the vicious circle between hypomagnesaemia and insulin resistance might explain the inverse relationship between daily magnesium intake and (type 2) diabetes mellitus [77,78].

The underlying mechanisms leading to obesity and metabolic syndrome, not only type 2 diabetes, are intertwined and gravitate around the involvement of magnesium in inflammation (in association with the mentioned insulin effects), which is why the nutrient has been analysed in various other metabolic and inflammatory conditions as well [79–81]. It is notable that magnesium deficiency decreases both extracellular and intracellular magnesium, leading to higher calcium influx and increased cytokine release, increased oxidative stress, and an increase in C-reactive protein [82,83].

Apart from low magnesium intake, sources and absorption also play a role in magnesium adequacy. In people with obesity, magnesium often comes from imbalanced diets with poor nutrient content but high in calories. This aspect might explain the connection between metabolic syndrome and a high intake of the micronutrient originating mostly from grains and potato, as mentioned prior [60] in contrast with the majority of all the other studies. Impaired absorption caused by gut inflammation and high calcium-to-magnesium ratio supplementation contribute to the altered magnesium balance in obese populations [84,85]. Collaterally, we mention a similar pattern with respect to vitamin D metabolism, where a higher cholecalciferol dose of replacement is required in patients diagnosed with obesity who are prone to vitamin D deficiency, an aspect that should be noted amid daily practice [86,87].

### 4.1.2. Cardiovascular Insights

Relationships have been identified between cardiovascular elements and daily magnesium intake, too, including high blood pressure (with or without metabolic syndrome diagnosis). A large study in which the relationship between the daily magnesium intake and blood pressure was investigated was performed by Han et al. [28] on 24,171 subjects. The lowest odds ratio of hypertension was found in the highest quintile of micronutrient intake (Q5 corresponded to 452 mg/day) [28]. Also, a large cross-sectional cohort study

regarding the impact of this dietary determinant on the blood pressure was performed by Cheteu Wabo et al. [88] on 16,684 subjects. This study established a connection between a low dietary magnesium intake and hypertension, finding that hypertensive subjects had a lower intake compared with subjects without high blood pressure (260.06 $\pm$ 1.91 versus 279.6 $\pm$ 2.22 mg/day). Moreover, the findings indicated a connection between magnesium and calcium intake since in the overall studied population and in male subjects, the lowest odds ratios of hypertension were found in individuals whose intake met the RDA for calcium and exceeded the RDA for magnesium. In women, hypertension had the lowest odds ratio in subjects with intake above the RDA both for calcium and for magnesium. A high intake of calcium ($\geq$1036 mg/day) and a high intake of magnesium ($\geq$322 mg/day) also reduced the odds ratio of high blood pressure [88]. The already mentioned study by Jiao et al. [59] also reported a lower hazard ratio in higher quantiles of daily magnesium intake compared with the lowest quantile (of less than 225.89 mg magnesium per day), but only with a statistically significant trend line ($p = 0.08$) [59]. Moreover, an increased risk of hypertension in subjects with a daily magnesium intake of less than 200 mg was also found in a large prospective cohort of 14,057 subjects who were followed for over 9 years [89].

Of note, apart from the distinct influence of daily magnesium intake, high blood pressure could be the effect of other nutrients including sodium and potassium. Some data showed that not only was daily dietary magnesium intake lower in patients with hypertension, but also that the potassium-to-magnesium ratio was negatively associated with the risk of high blood pressure when comparing the group with potassium and magnesium intake below the RDA with the group with potassium and magnesium intake above the RDA [90].

Additionally, less homogenous results highlighted a potential connection between lower magnesium intake and the risk of arterial disease. For example, Wu et al. [91] found reduced nutrient intake in subjects diagnosed with peripheral artery disease (of 244.89 $\pm$ 6.31 versus 288.30 $\pm$ 3.66 mg/day). This was an NHANES analysis between 1999 and 2004 that was published in 2023 [91]. However, Vermeulen et al. [92] did not identify a statistically significant association between the dietary determinant and vascular functional and structural markers [92]. The overall risks of cardiovascular and atherosclerotic cardiovascular disease were investigated by Pickering et al. [93] in a cohort of 2362. The study from 2021 found that these mentioned risks were reduced by a daily intake over 320 mg/day. Moreover, the risk was also reduced by a magnesium intake of more than 240 mg/day in individuals with a low daily sodium intake (of less than 2.5 g/day) [93].

In terms of interventional studies, the mentioned study of Afitska et al. [51] found that daily magnesium supplementation of 400 mg reduced both systolic and diastolic blood pressure in subjects with metabolic syndrome [51]. Another cohort found an improvement in systolic blood pressure after supplementation with 360 mg per day, but only in subjects with a blood pressure value higher than 132 mm Hg at baseline, but no analysis of the effect size was conducted due to the small sample size [94].

There are multiple underlying magnesium-related components of the pathogenic traits of hypertension and they go beyond magnesium acting as an antagonist to calcium. At the vascular smooth muscle cell level, both intracellular and extracellular magnesium impact contractility. By neutralizing the negative charge of the outer layer of the membrane leading to an increased depolarization threshold and by binding to calcium channels, extracellular magnesium reduces the influx of calcium. The intracellular magnesium not only reduces calcium influx, but also promotes calcium sequestration in the endoplasmic reticulum and competes with calcium for binding to contractile proteins [95–97].

Additionally, magnesium is involved in the hormonal pathways by reducing aldosterone secretion as well modulating the sensitivity to catecholamines [98,99]. Also, a decrease in the nutrient content contributes to arterial stiffness and increases endothelial dysfunction and oxidative stress, all of which have been linked to hypertension [100,101]. Whether magnesium supplementation improves the hypertension profile remains an open issue since a panel of heterogeneous results has been found so far [102–105]. Larger studies

are still needed to specifically quantify the benefit of magnesium supplementation and to determine the best supplementation dose in order to achieve an optimum effect with regard to blood pressure.

*4.2. Other Clinical Elements with Magnesium-Intake-Related Considerations*

As mentioned, numerous other metabolic ailments, including those that are related to kidney status, skeleton health, and mineral metabolism, have been associated with daily magnesium intake, either as potential pathogenic contributors or as part of lifestyle interventions under certain circumstances.

4.2.1. Interplay between Magnesium Intake and Kidney Status

According to some data, many patients diagnosed with end-stage renal disease and under chronic dialysis have inappropriate daily magnesium intake, especially since these are associated with a low-sodium diet. Another perspective showcased the correlation between low dietary nutrient intake and increased risk of developing chronic renal impairment [106–108]. This aspect might be explained by the calcification of the vascular smooth muscle cells as promoted by magnesium deficiency in addition to the higher risk of atherosclerosis, endothelial dysfunction, and metabolic syndrome components including type 2 diabetes [109–112]. At the other end of the spectrum, a narrow frame of intervention should be taken into consideration since hypermagnesaemia is also detrimental in chronic kidney disease as it may contribute to adynamic bone disease and osteomalacic renal osteodystrophy [113]. The pathway that leads from an adequate dietary magnesium intake to a better outcome of diabetic nephropathy, though incompletely understood, also includes the antioxidant properties of magnesium and its involvement in reducing inflammation and oxidative stress, as well as the role in blood pressure control [114].

Clinical data, as we found collaterally across our search, revealed the largest study investigating the relationship between daily magnesium intake and kidney disease (N = 19,271 subjects) that pinpointed the fact not only that individuals with kidney stones had lower magnesium intake per day compared to those with kidney stones (295.4 mg/day versus 309.6 mg/day), but also a lower odds ratio of 0.70 (95% CI between 0.52 and 0.93) of having kidney stones was associated with the highest daily magnesium intake quartile (of 379 to 2725 mg/day) compared with the lowest quartile (of ≤240 mg/day) [115]. Moreover, a small-sample-size cross-sectional study (N = 83 people with urolithiasis) found a statistically significant negative correlation between intake of this nutrient and net endogenous acid production (r = −0.233, *p* = 0.03), suggesting the potential protective role of magnesium-rich foods against urolithiasis [116].

The main pathogenic connection between magnesium and kidney stones is the formation of calcium oxalate. Due to binding to oxalate, higher urinary magnesium leads to a reduced concentration of urinary calcium oxalate. In addition, in the intestines, magnesium reduces the absorption of oxalate. These processes lead to less crystal formation. Magnesium not only reduces the formation of these crystals, but also their size and growth [117–119]. Even though magnesium intake was associated with kidney stones, there is a lack of recent statistical evidence regarding the benefits of supplementation. A question worth asking would be whether supplementation with magnesium might substantially reduce the incidence of the kidney stones. Further research is needed to understand the best daily magnesium intake doses that are protective against chronic kidney disease and with the safest profile of side effects.

4.2.2. Dietary Magnesium Amid the Diagnosis of Osteoporosis

Magnesium deficiency may promote osteoporosis through multiple direct and indirect pathways; for example, it disturbs the balance between osteoclasts and osteoblasts favouring bone loss, particularly at the level of cortical bone. Bone quality is also reduced by the formation of large hydroxyapatite crystals [120,121]. The lowered PTH secretion caused by magnesium deficiency contributes to a decrease in bone formation. Moreover,

by promoting inflammation and oxidative stress, magnesium deficit may elevate bone resorption [122,123]. The effects of magnesium are not limited to the bone, as findings suggest that an adequate magnesium intake or supplementation may prevent sarcopenia to a certain level as well [123,124].

In particular, as a collateral finding to our main results, one large study found statistically significantly lower (263.1 ± 114.3 mg/day) magnesium intake in subjects with osteoporosis (N = 14,566 subjects with osteoporosis; 82.2% were females) compared with subjects without osteoporosis (304.5 ± 126.5 mg/day, *p* < 0.001). The same study also found an increased risk of osteoporosis associated with a magnesium depletion score higher than 3 [odds ratio of 1.785 (95% CI between 1.544 and 2.064), *p* < 0.001] [125]. Another cohort published in 20202 on 194 young females showed that women with insufficient magnesium intake had lower bone mineral density (*p* = 0.03) [126]. A longitudinal study on 144 females found a negative correlation between the bone resorption marker C-Telopeptide of type I collagen and daily magnesium intake (r = −0.21, *p* = 0.02) [127]. In men, a prospective study on 61,025 subjects found a lower daily magnesium intake in men with osteoporotic fractures (of 312 ± 78.3 mg/day) versus controls (324 ± 83.6 mg/day; *p* = 0.001). However, higher intakes of magnesium (such as >450 mg/day versus ≤250 mg/day) were associated with an increased risk of osteoporotic fractures [odds ratio of 2.12 (95% CI between 1.15 and 3.91) versus 2.21 (95% CI between 1.08 and 4.50), respectively] [128]. In addition, dietary magnesium may influence the association between osteoporosis and other comorbidities, such as, as already mentioned, the presence of kidney stones. An intake higher than 350 mg/day was associated with lower odds of osteoporosis or osteopenia and kidney stone comorbidities [odds ratio of 0.639 (95% CI between 0.472 and 0.866), *p* = 0.004] [129].

In clinical practice, however, magnesium supplementation is yet to be proven effective in improving bone health, as current studies show inconclusive results [130]. Larger trials are needed to explore the adequate dose and quantify its benefit. Further research should focus on the impact of magnesium on bone quality and explore a possible relationship with trabecular bone score considering the influence of type 2 diabetes.

### 4.2.3. Magnesium and Mineral Metabolism: PTH Crossways

By binding to the calcium sensor receptor, magnesium inhibits PTH secretion, albeit less potently than calcium. It is worth noting that even though hypomagnesaemia stimulates PTH secretion, severe hypomagnesaemia exerts an inhibitory effect. On the other hand, low intracellular magnesium alters this receptor signalling and decreases hormone secretion [131–133]. In the small intestine, magnesium is absorbed through both para-cellular and trans-cellular mechanisms [134]. In an animal model, trans-cellular absorption was reversed by the magnesium supplementation [135]. This leads to the question whether magnesium supplementation may benefit patients with hyperparathyroidism. Magnesium deficiency in patients with hyperparathyroidism was found in a quarter to a third of patients [136,137]. This high prevalence appears to bring a series of consequences. First of all, low serum magnesium was associated with symptomatic disease [136,137]. Secondarily, similarly to the general population, hypomagnesaemia increased the risk of kidney stones in patients with PTH-related ailments too [138]. It seems that not only low serum magnesium increases the risk of kidney stones, but also hypomagnesaemia [139]. Hence, the complex interplay between magnesium status and PTH profile, including in pathological conditions such as parathyroid tumour, might be an important, yet incompletely understood, contributor to the overall clinical picture.

Our comprehensive review highlighted across 30 studies an important update with regard to daily magnesium intake and glucose profile as part of the modern medical era that associates prompt lifestyle interventions with a better outcome. As limits, we mention the study's design in terms of narrative review, an option that allowed us a more flexible approach since the methods of the cited papers largely varied. Further trials will pinpoint the areas of discordant or rather heterogeneous results as we have mentioned prior.

## 5. Conclusions

Across the data highlighted from the most recent 30 clinical studies, we may conclude that diabetic patients are associated with a lower daily magnesium intake. Furthermore, this may be correlated with poorer disease control and an increased risk of complications such as retinopathy and nephropathy and even with elevated mortality indices to a certain level. In the general population, a heterogeneous spectrum of data was identified in order to pinpoint a higher risk of displaying abnormal HOMA-IR and other glucose profile elements if the dietary nutrient was deficient. Finally, in terms of magnesium supplementation, while seven randomized controlled studies showed promising results, they are not generally applicable since the sample sizes of the mentioned cohorts included than less than 100 patients per study. These clinical aspects are essential from a multidisciplinary perspective and further trials are mandatory to address the current areas of discordant results.

**Author Contributions:** Conceptualization, A.-M.G., M.-L.C., C.N., M.-M.G., B.-A.S., M.S., F.L.P. and M.C.; methodology, A.-M.G., M.-L.C., M.-M.G., B.-A.S., M.S., F.L.P. and M.C.; software, A.-M.G., M.-L.C., C.N., B.-A.S., M.S., F.L.P. and M.C.; validation, A.-M.G., M.-L.C., C.N., M.-M.G., B.-A.S. and M.C.; formal analysis, A.-M.G., M.-L.C., C.N., M.-M.G., M.S., F.L.P. and M.C.; investigation, A.-M.G., M.-L.C., C.N., M.S., F.L.P. and M.C.; resources, A.-M.G., M.-L.C., C.N., M.-M.G., B.-A.S. and M.C.; data curation, A.-M.G., C.N., M.-M.G., B.-A.S., M.S., F.L.P. and M.C.; writing—original draft preparation, A.-M.G. and M.C.; writing—review and editing, A.-M.G., M.-L.C., C.N., M.-M.G., M.S., F.L.P. and M.C.; visualization, A.-M.G., M.-L.C., C.N., M.-M.G., B.-A.S., M.S., F.L.P. and M.C.; supervision, A.-M.G., M.-L.C., C.N. and M.C.; project administration, M.-L.C., C.N. and M.C.; funding acquisition, M.C. All authors have read and agreed to the published version of the manuscript.

**Funding:** This research received no external funding.

**Institutional Review Board Statement:** Not applicable.

**Informed Consent Statement:** Not applicable.

**Data Availability Statement:** Not applicable.

**Acknowledgments:** This is part of PhD research of the PhD Doctoral School of "Carola Davila" University of Medicine and Pharmacy, entitled "Primary hyperparathyroidism: cardio-metabolic, osseous and surgical aspects"—28374/2.10.2023.

**Conflicts of Interest:** The authors declare no conflicts of interest.

## Abbreviations

| | |
|---|---|
| ATP | adenosine triphosphate |
| cAMP | cyclic adenosine monophosphate |
| CI | confidence interval |
| EEFSA | European Food Safety Association |
| HbA1c | glycated haemoglobin A1c |
| HOMA-IR | Homeostatic Model Assessment for Insulin Resistance |
| NAHSIT | Nutrition and Health Survey in Taiwan |
| NHANES | National Health and Nutrition Examination Surveys (NHANES), |
| NIPAL1 | Nuclear Interaction Partner of Alkaline Phosphatase-like domain containing 1 |
| PTH | parathyroid hormone |
| RDA | recommended dietary allowance |
| QUICKI | quantitative insulin sensitivity check index |

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
