# Peer review of "Inquiry of the Metabolic Traits in Relationship with Daily Magnesium Intake: Focus on Type 2 Diabetic Population"

_clinpract, doi:10.3390/clinpract14040107_

Round 1
Reviewer 1 Report
Comments and Suggestions for Authors
Hermosillo, June 7th, 2024
Inquiry of the metabolic traits in relationship with daily magnesium intake: focus on type 2 diabetic population
Magnesium represents an essential nutrient with a wide area of physiological roles, including being a cofactor in over 600 enzymatic reactions involved in the synthesis of proteins and nucleic acids, DNA repair, neuromuscular functions, neuronal transmission, cardiac rhythm regulation, the modulation of metabolic pathways as well as acting as natural blocker for the calcium channels. The European Food Safety Association (EFSA) recommends a daily magnesium intake of 350 mg for men, and 300 mg for women, while for US the recommendation is slightly higher (420 mg/day for males and 320 mg/day for females) . In humans, the most important dietary sources include plant-based foods rich in magnesium (such as whole grains, green vegetables, nuts, seeds) in addition to sources with a lower magnesium content (for example, legumes, fruit, meat, fish, and dairy). When it comes to (oral) magnesium supplementation, it should be noted that the bioavailability of different supplements largely varies, with organic formulations having a higher bioavailability versus inorganic products [9]. In spite of a wide variety of dietary sources, magnesium deficiency is quite common in recent 80 years, and it is linked to numerous conditions (with pathogenic loops being more or less understood so far) including cardiovascular disease, type 2 diabetes mellitus, gastrointestinal conditions, chronic kidney disease, and osteoarticular disease, including low bone mineral density and even some parathyroid conditions.
The topic presented by the authors is very interesting and very worrying, especially today there are many diabetic patients who suffer and do not know what to do and I believe. That this type of work must exist to solve this disease.
After reading the complete manuscript, I have observed that the literature consulted for this manuscript is well updated.
In relation to these observations, I have a few questions:
1. How important is the interaction of parathyroid hormone (PTH), calcium and the balance between intracellular and extracellular magnesium content?
2. How are serum magnesium levels regulated in the human body?
3. After analyzing the clinical studies compiled by you, what doses of magnesium and zinc would you recommend for diabetic and non-diabetic patients?
4. Do you think that supplementation of minerals such as magnesium and zinc is good for the treatment of type 1 and type 2 diabetics? What would be your recommendations?
Author Response
Response to Review 1 Comments
Dear Reviewer,
Thank you very much for your time and your effort to review our manuscript.
We are very grateful for providing your valuable feedback on the article.
Here is our response and related amendment that has been made in the manuscript according to your review (marked in yellow color).
Inquiry of the metabolic traits in relationship with daily magnesium intake: focus on type 2 diabetic population
Magnesium represents an essential nutrient with a wide area of physiological roles, including being a cofactor in over 600 enzymatic reactions involved in the synthesis of proteins and nucleic acids, DNA repair, neuromuscular functions, neuronal transmission, cardiac rhythm regulation, the modulation of metabolic pathways as well as acting as natural blocker for the calcium channels. The European Food Safety Association (EFSA) recommends a daily magnesium intake of 350 mg for men and 300 mg for women, while for US the recommendation is slightly higher (420 mg/day for males and 320 mg/day for females). In humans, the most important dietary sources include plant-based foods rich in magnesium (such as whole grains, green vegetables, nuts, seeds) in addition to sources with a lower magnesium content (for example, legumes, fruit, meat, fish, and dairy). When it comes to (oral) magnesium supplementation, it should be noted that the bioavailability of different supplements largely varies, with organic formulations having a higher bioavailability versus inorganic products [9]. In spite of a wide variety of dietary sources, magnesium deficiency is quite common in recent 80 years, and it is linked to numerous conditions (with pathogenic loops being more or less understood so far) including cardiovascular disease, type 2 diabetes mellitus, gastrointestinal conditions, chronic kidney disease, and osteoarticular disease, including low bone mineral density and even some parathyroid conditions.
The topic presented by the authors is very interesting and very worrying, especially today there are many diabetic patients who suffer and do not know what to do and I believe. that this type of work must exist to solve this disease.
Thank you very much. We really appreciate it!
After reading the complete manuscript, I have observed that the literature consulted for this manuscript is well updated.
Thank you very much.
In relation to these observations, I have a few questions:
- How important is the interaction of parathyroid hormone (PTH), calcium and the balance between intracellular and extracellular magnesium content?
Thank you very much. PTH and calcium-related normal and abnormal profile should be analyzed in relationship with magnesium content as we mentioned the followings, noting that PTH stands as a modular of the nutrient reabsorption by elevating its influx into the distal tubule cells via protein kinase A and C pathways:
..”magnesium deficiency is quite common in recent years, and it is linked to numerous conditions (with pathogenic loops being more or less understood so far) including… even some parathyroid conditions”..
“The relationship between the daily magnesium intake and its serum levels is regulated by complex mechanisms also including the balance between nutrient absorption and excretion, the adjustment of its bone storage, the interaction with parathyroid hormone (PTH) and calcium (as essential contributors of the mineral metabolism), and the equilibrium between intracellular and extracellular magnesium content”
…”adults whereas a negative correlation between magnesium intake per day and serum parathyroid hormone and 25-hydroxyvitamin D levels was confirmed”…
“This leads to the question whether magnesium supplementation may benefit patients with hyperparathyroidism. Magnesium deficiency in patients with hyperparathyroidism was found in a quarter to a third of patients [136-137]. This high prevalence appears to bring a series of consequences. First of all, low serum magnesium was associated with symptomatic disease [136-137]. Secondarily, similarly to the general population, hypomagnesemia increased the risk of kidney stones in patients with PTH-related ailment, too [138]. It seems that not only low serum magnesium increases the risk of kidney stones, but also hypomagnesemia [139]. Hence, the complex interplay between magnesium status and PTH profile, including in pathological conditions such as parathyroid tumour might be an important, yet, incompletely understood, contributor to the overall clinical picture.”
Thank you
- How are serum magnesium levels regulated in the human body?
Thank you very much. We added the followings:
“Magnesium-related physiological profile include a balance between the dietary intake, nutrient absorption at distal small intestinal level, respectively, its renal excretion and reabsorption versus micronutrient storage into the bones (representing 50-60% of the total body content) and soft tissues such as the muscle, etc. Additional regulation is provided by passive para-cellular and trans-cellular transport at cecal and colonic level. The renal excretion is regulated by the serum levels despite the fact than less than 1% is found here.”
Thank you
- After analyzing the clinical studies compiled by you, what doses of magnesium and zinc would you recommend for diabetic and non-diabetic patients?
Thank you very. The doses of magnesium were analyzed in diabetic population versus controls as mentioned within Table 6 and additional information across the cited randomized studies. The supplementation was done with the following regimes:
Supplementation with 150 mL Mg gluconate (= 360 mg Mg)
Supplementation with 250 mg Mg oxide
Supplementation with 350 mg Mg
Supplementation with 250 mg Mg oxide + 150 mg Zn sulfate
Supplementation with 400 mg Mg citrate
Supplementation with 2.25 g Mg citrate (= 360 mg Mg)
The data with concern to zinc are out of the scope with respect to the present work
Thank you
- Do you think that supplementation of minerals such as magnesium and zinc is good for the treatment of type 1 and type 2 diabetics? What would be your recommendations?
Thank you very much. Indeed, as mentioned across the paper, the magnesium supplementation helps type 2 diabetic population, but does not replace the standard anti-diabetic care by itself. The spectrum of doses is still heterogeneous, as mentioned, for instance:
…”both fasting plasma glucose and insulin decreased in patients with diabetes mellitus who received supplementation with 250 mg of magnesium oxide and 150 mg of zinc sulphate, as revealed by a double-blind randomized controlled trial performed by Hamedifard et al….”
…”HOMA-IR improved by supplementation with 350 mg of magnesium daily from 440 mL of balanced deep-sea water in a randomized double-blind crossover trial performed by Ham et al…”
…”A lowering in HbA1c levels in people with metabolic syndrome (–0.28 ± 0.27, p=0.0036) was also reported following a supplementation with 400 mg magnesium daily
The data with concern to zinc are out of the scope with respect to the present work…”
Thank you
Thank you very much.

Reviewer 2 Report
Comments and Suggestions for Authors
Magnesium intake and its level participated in the glucose metabolism, the importance of which was underestimated. It is good to find someone to summarise the literature in such an interesting topic.
1. It the Mg levels as well as potassium levels can be mediators for the relationship between Mg intake and glucose metabolic impairment. It is necessary to illustrate a DAG for the potential causal link.
2. Please list cross-sectional study and cohort studies separately.
3. Please consider the Mg intake and levels in long-term outcomes.
Comments on the Quality of English LanguageJust OK
Author Response
Response to Review 2 Comments
Dear Reviewer,
Thank you very much for your time and your effort to review our manuscript.
We are very grateful for your insightful comments and observations, also, for providing your valuable feedback on the article.
Here is a point-by-point response and related amendments that have been made in the manuscript according to your review (marked in yellow color).
Magnesium intake and its level participated in the glucose metabolism, the importance of which was underestimated. It is good to find someone to summarize the literature in such an interesting topic.
Thank you very much. We really appreciate it!
- It the Mg levels as well as potassium levels can be mediators for the relationship between Mg intake and glucose metabolic impairment. It is necessary to illustrate a DAG for the potential causal link.
Thank you very much for your recommendations. We introduced Figure 1 accordingly:
”Figure 1. Magnesium regulates by being an enzyme co-factor insulin secretion and receptor effects as well as oxidative stress and inflammatory profile, all of which might be impaired amid the detection of type 2 diabetes [22-25]”
Thank you
- Please list cross-sectional study and cohort studies separately.
Thank you very much. A distinct analysis with respect to the specific types of the studies that we could find according to our methods was out of our scope since this is a non-systematic review. We respectfully mention that the data regarding the studies designs were provided for each study within the six tables, thus, by splicing them, we do not intend to complicate the data presentation. Thank you
- Please consider the Mg intake and levels in long-term outcomes.
Thank you very much. This is an interesting point. However, all the data available according to our methods with respect to this specific interventional section have been introduced in Table 6 and associated main text. The level of statistical evidence remains rather limited and long term outcome, despite being very useful, did not represent the endpoint of the mentioned randomized interventional trials as we could find them. According to your observations, we added: “Further analysis of long-term outcomes in the field of magnesium supplementation is mandatory.”
Thank you
Comments on the Quality of English Language: Just OK
Thank you.
Thank you very much,

Reviewer 3 Report
Comments and Suggestions for Authors
This review article is very interesting. Expanding the perspective on disease management can be of great help in managing diabetic and pre-diabetic patients. However, there were several flaws in publishing this article. I think it will be a good study if revised.
1. Methodology section should be separated with introduction. What kind of literature search engine was used? How many people participated in the study search? Why did you exclude experimental studies?
2. Line 100, it was reported that experimental studies were exclude. However, from line 313, all literatures from 50th to 54th that you analyzed were experimental studies. How do you explain this?
3. You used so many explanations in bracket in the middle of sentences. It reduces readability. Please revise them.
4. Number 2 had no title. “2. Results(?)”
5. What are the differences in literatures between results and discussion sections? Aren’t cited literatures from 26 to 54 research data? However, the discussion section uses other literature to discuss obesity and metabolic syndrome, cardiovascular insights, and other clinical factors, etc. It is very confusing.
6. It will be a similar context of number 4. How did you derive the subthemes in the discussion? For example, where does the subtopic “magnesium intake and metabolic features” in the discussion come from? Are the results columns from “magnesium intake in diabetic population” and “magnesium intake and the risk of diabetes mellitus” in the result section? Linking the results to the discussion will help readers understand your study.
Comments on the Quality of English LanguageMinor editing of English language required
Author Response
Response to Review 3 Comments
Dear Reviewer,
Thank you very much for your time and your effort to review our manuscript.
We are very grateful for your insightful comments and observations, also, for providing your valuable feedback on the article.
Here is a point-by-point response and related amendments that have been made in the manuscript according to your review (marked in yellow color).
This review article is very interesting. Expanding the perspective on disease management can be of great help in managing diabetic and pre-diabetic patients. However, there were several flaws in publishing this article. I think it will be a good study if revised.
Thank you very much
- Methodology section should be separated with introduction. What kind of literature search engine was used? How many people participated in the study search? Why did you exclude experimental studies?
Thank you very much. We separated the Methods from Introduction. This was a PubMed search. The contribution of each author is specified at the end of the paper according to MDPI rules. This is not a study, but a comprehensive. We corrected as followings: “we excluded experimental studies in non-type 2 diabetes”. Thank you very much for pointing this aspect.
- Line 100, it was reported that experimental studies were exclude. However, from line 313, all literatures from 50thto 54th that you analyzed were experimental studies. How do you explain this?
Thank you very. We prior specified at point number 1 the correction of exclusion with respect to experimental studies other than type 2 diabetic populations. Thank you
- You used so many explanations in bracket in the middle of sentences. It reduces readability. Please revise them.
Thank you very much. We revised and excluded them, if feasible. Thank you
- Number 2 had no title. “2. Results(?)”
Thank you very much. We corrected the title of each section in terms of name and section number. Thank you
- What are the differences in literatures between results and discussion sections? Aren’t cited literatures from 26 to 54 research data? However, the discussion section uses other literature to discuss obesity and metabolic syndrome, cardiovascular insights, and other clinical factors, etc. It is very confusing.
Thank you very much. The Discussion represents an expansion of the topic from Results section in terms of magnesium and type 2 diabetes mellitus. This is because one patient might suffer as well from obesity, metabolic syndrome with/without parathyroid conditions in addition to the diagnosis of diabetes and magnesium profile-related analysis might play a role. The Discussion section introduces further implications of the analyzed studies across Results section. This includes either the same studies from Results that highlighted some aspects with respect to a larger endocrine-metabolic frame, either other studies were identified across our search that do not fulfill the criteria for Results, but were consider important to be discussed/mentioned in this section. Discussion section opens new doors in the field of type 2 diabetes and magnesium, new domains and connections that worth to be mentioned in relationship to the specific scope of the present work. Thank you
- It will be a similar context of number 4. How did you derive the subthemes in the discussion? For example, where does the subtopic “magnesium intake and metabolic features” in the discussion come from? Are the results columns from “magnesium intake in diabetic population” and “magnesium intake and the risk of diabetes mellitus” in the result section? Linking the results to the discussion will help readers understand your study.
Thank you very much. These sections from Results are introduced within the mentioned subsections and, taking into account that there is a higher risk of metabolic syndrome in one patient confirmed with type 2 diabetes, this frame of understanding is important to be discussed. We specified these aspects in several parts, for instance:
“The nutrient intake has been found in correlation not only to the glucose profile, but, also, with other cardio-metabolic features, thus the importance of addressing this topic and its complexity.”
“To summarize, a lower magnesium intake was linked to an elevated prevalence of diabetes mellitus, obesity, and metabolic syndrome, in most studies. The adherence to daily recommendations was decreased in diabetic people with or without obesity. However, a large variability in assessing the adherence to optimum nutrient doses was noted and this might come as a potential bias.”
“Whether specific tailoring of daily magnesium intake recommendations to subgroups in terms of age and sex should be differently approached with concern to each metabolic component is yet an open issue.”
“The underlying mechanisms leading to obesity and metabolic syndrome, not only type 2 diabetes, are intertwined and gravitate around the involvement of magnesium in inflammation..”
Thank you.
Comments on the Quality of English Language: Minor editing of English language required
Thank you. We corrected it.
Thank you very much,

Round 2
Reviewer 3 Report
Comments and Suggestions for Authors
Thank you for your efforts to modify your manuscript.
1. Still, there seems to be some incongruity in the transition from results to discussion. It would be better to match the title and content of the results with the discussion.
2. why did you exclude the experimental study in non-type 2 diabetes?
3. And, several typo were shown, which should be corrected. For example, line 297, p<.00 would be typo. Please check your manuscript with a strict eye.
Comments on the Quality of English Language
Moderate editing of English language required
Author Response
Response to Review 3 Comments (Round 2)
Dear Reviewer,
Thank you very much for your time and your effort to review our manuscript for the second time.
We are very grateful for providing your valuable feedback on the article.
Here is our response and related amendment that has been made in the manuscript according to your second review (marked in yellow color).
Thank you for your efforts to modify your manuscript.
Thank you very much. We really appreciate all the observations, recommendations, and suggestions.
- Still, there seems to be some incongruity in the transition from results to discussion. It would be better to match the title and content of the results with the discussion.
Thank you very much. Particularly, the issue of type 2 diabetes mellitus cannot be regarded as an isolated ailment and the entire metabolic and cardiovascular panel should be analyzed in one patient or one population, including in relationship with the magnesium status interplay, specifically, the nutrient intake and its deficiency. This should help the clinicians to have a multidisciplinary complex approach and to assess the clinical management across an in-depth perspective which can only help the patients to achieve an overall better outcome. For this, we looked at the tidily connected data of the metabolic syndrome and cardiovascular risk with regard to magnesium intake. We added a figure to highlight this connection and make it clearer. Thank you
- Why did you exclude the experimental study in non-type 2 diabetes?
Thank you very much. We excluded them because of the followings:
- This is out of the scope of the present work as shown within the title.
- The other types of diabetes such as type 1 (insulin dependent) might come with additional bias due to the interplay between the insulin profile and underlying signal transduction pathways amid insulin therapy and the magnesium status within the human body.
- The other types of diabetes mellitus in terms of endocrine (secondary) diabetes as seen, for instance, in acromegaly or Cushing’s syndrome, are out of the aim of the present work and they bring other confounding factors in this interplay for instance, according to the way that magnesium might interfere with other hormones’ production, secretion and effects such as growth hormone and cortisol. For these particular purposes, we only focused on type 2 diabetes which, from the epidemiologic perspective, represents the most important type all over the world due to a very high and increasing prevalence at all ages, particularly in adults.
Thank you
- And, several typo were shown, which should be corrected. For example, line 297, p<.00 would be typo. Please check your manuscript with a strict eye.
Thank you very much. We corrected it.
Comments on the Quality of English Language
Moderate editing of English language required.
Thank you very much. We revised the English language.
Thank you very much.
